# Production of Long Fermentation Bread with Jabuticaba Peel Flour Added: Technological and Functional Aspects and Impact on Glycemic and Insulinemic Responses

**DOI:** 10.3390/foods13182878

**Published:** 2024-09-11

**Authors:** Miriam Regina Canesin Takemura, Juliana Silva da Graça, Marianna Miranda Furtado, Marcella Camargo Marques, Anderson S. Sant’Ana, Mário Roberto Maróstica Junior, Lilian Regina Barros Mariutti, Bruno Geloneze, Cinthia Baú Betim Cazarin

**Affiliations:** 1Department of Food Science and Nutrition, School of Food Engineering, Universidade Estadual de Campinas, Campinas CEP: 13083-862, SP, Brazil; miriamcanesin@gmail.com (M.R.C.T.); julianagraa@hotmail.com (J.S.d.G.); marianna.mirandafurtado@gmail.com (M.M.F.); marques.marcella@gmail.com (M.C.M.); and@unicamp.br (A.S.S.); mmarosti@unicamp.br (M.R.M.J.); lilianma@unicamp.br (L.R.B.M.); 2Obesity and Comorbities Research Center (OCRC), Universidade Estadual de Campinas, Campinas CEP: 13083-864, SP, Brazil; bgeloneze@gmail.com

**Keywords:** anthocyanins, bioactive compounds, nutritional composition, Brazilian native fruit

## Abstract

The consumption of jabuticaba (*Plinia cauliflora*), a native Brazilian fruit, has shown promising results concerning some metabolic disorders. Therefore, studying it may aid in the development of products capable of preventing or delaying pathological conditions. The objective of the present study was to formulate a functional sourdough bread with the inclusion of jabuticaba peel flour (JPF) and to evaluate the effect on the postprandial response. The proximate composition of the JPF and bread, the stability of the antioxidant compounds after baking, and the functional activity in a clinical trial were carried out to develop the product and guarantee its quality. JPF increased the fiber content of the bread in comparison to the control from 1.0 g/100 g to 2.3–2.9 g/100 g. Also, the antioxidant capacity increased 1.35 to 3.53 times by adding JPF to the bread, as well as the total reducing capacity (1.56 to 2.67 times). The bread’s shelf life was guaranteed for seven days. In the clinical trial, the consumption of control bread resulted in a glycemia peak at 30 min, remaining high until 45 min; for the JPF bread, we noticed a less prominent peak at 45 min and a decrease with less inflection until 180 min. The serum antioxidant capacity of the individuals increased after the 3 h. Although no statistical difference was observed between the groups in the satiety profile, JPF bread presented higher scores after 60 min. Furthermore, a lessened desire to eat other foods and hunger was observed after consuming JPF bread. The inclusion of JPF in the bread manufacturing process promoted a longer shelf life and increased nutritional quality.

## 1. Introduction

Bread is an ancient food that is part of many civilizations’ food culture, being present at critical moments in the history of humanity. Although bread is the basis of the diet of many cultures, its consumption in some situations is questioned due to its composition, either because it is a primary source of carbohydrates or because of the use of additives in its formulation.

In this sense, artisanal breadmaking has sought to diversify products with an emphasis on the healthiness attribute, with the inclusion of whole grains in formulations to add nutritional value, to reduce fats, oils, sugar, and salt, in an attempt to prevent bread consumption from aiding the development of some pathologies [1]. Artisanal breadmaking has grown strongly in recent years; it uses rudimentary methods of bread making involving sourdough or natural yeast, also known as levain, as an alternative to the usual additives. This can increase shelf life and nutritional value and contribute to the development of a peculiar flavor [2].

Traditional sourdough is made from a mixture of flour and water, a method that, due to time and temperature, favors fermentation by lactic acid bacteria (LABs) and the yeast native to wheat flour. After several biochemical processes, the production of organic acids acidifies the medium and induces proteolysis, enzyme and antifungal compound synthesis, and the release of exopolysaccharides [3]. Considering the regionalities and particularities of artisanal baking techniques, we can classify sourdough into four types, differing in the method of introducing ingredients, starter yeast, and time to obtain the final product, namely: Type I, exclusive fermentation of autochthonous and environmental flour microbiota (bacteria and yeast), obtained at room temperature (24 °C) for around 5 to 10 days, which requires successive back-slopping to stimulate microbiota activity and introduce nutrients to the substrate; Type II, characterized by the addition of starter cultures to the substrate consisting of flour and water with a controlled temperature of 30 °C, being a quick process requiring only one fermentation cycle before use, which takes place over 15 to 24 h; Type III, obtained from the dehydration of type II and the most widespread for “commercial use”, can be active or inactive, used not precisely as a yeast but rather as a sensory improver providing sensory properties close to type I; Type IV, the most prolonged and expensive process, which consists of a mixture of types I and II—a starter culture is added, but the process continues with back-slopping and fermentation cycles [4,5]. According to Brandt [6], in addition to being traditionally used as a fermentation agent and improver, sourdough combined with commercial bakery yeast also allows for the replacement of additives, promoting clean labels. In this way, sourdough aims to improve the quality of bread and replace additives due to the transformations, reactions, and products generated during fermentation [7].

In recent years, the focus of studies has been on the use and inclusion of ingredients with high biological value, including grains, fibers, bioactive compounds, pigments, and natural preservatives from various sources. Using some by-products, mainly of plant origin, such as fruit peels, has great potential to become food ingredients intended for human consumption, mainly because they are a general source of these substances [8]. Furthermore, the use of by-products from the agricultural industry, in forms including bagasse, filter pulp, sieve residue, extraction cake, peels, seeds, and leaves, has attracted attention from the scientific community due to the large volume produced, their potential use as sources of nutraceutical/functional co-products, and the environmental impact associated with their disposal [9].

In this sense, the present study explored the potential of jabuticaba peel (*Plinia cauliflora*), a typical Brazilian fruit, to improve bread’s nutritional qualities. Its purplish-colored peel, due to the presence of anthocyanins, is also a source of soluble and insoluble fibers [10,11,12]. The presence of fermentable products (sugars) and acidic pH (phenolic acids) suggests that jabuticaba peel has the potential to be used as an adjuvant in the fermentation of sourdough and can add value to long-fermented bread because of the presence of fiber and phenolic compounds [13,14]. In this sense, the objective of the present study was to formulate a long-fermented functional artisanal bread with the addition of jabuticaba peel flour and to evaluate its effects on the postprandial response (glycemic and insulinemic responses) and serum antioxidant capacity.

## 2. Materials and Methods

### 2.1. Materials

The jabuticaba fruits (*Myrciaria jabuticaba* (Vell.) Berg.) were obtained from producers in Casa Branca/SP, Brazil, in November 2017. The other ingredients for making the bread were purchased in local supermarkets in Campinas/SP, Brazil. The activity of accessing Genetic Heritage/CTA was registered with SisGen under registration number A5665F9, in compliance with Law No. 13,123/2015 and its regulations.

The fruits were washed, sanitized in hypochlorite 1% (*v*/*v*) for 10 min, rinsed, manually pulped, and frozen at −18 °C. The peels were dried in a continuous airflow dryer at 50 °C for 5 h to obtain the dry peel. After drying, the peels were crushed in a blender until they formed a homogeneous mixture.

### 2.2. Bread Formulation

Three concentrations of JPF (10%, 7.5%, and 5%) were added to the control formulation of naturally fermented bread to evaluate technological and functional parameters (Table 1). The breads were prepared according to the direct mixing method and the slow fermentation step shown in Figure 1. For shelf life analyses, the breads were cooled for 1 h after baking until they reached room temperature (24 °C); they were then packaged in plastic bags.

### 2.3. Approximate Composition—Jabuticaba Peel Flour and Bread

The approximate composition of the bread was determined based on official methods. The moisture content of the pieces of bread was determined by oven drying at 70 °C until constant weight and protein (by the micro-Kjeldhal method) was achieved [15]. Lipids were quantified using the method of Bligh and Dyer [16]. Ash content was estimated by incineration of the organic material in a muffle furnace at 550 °C. The percentages of total, soluble, and insoluble dietary fiber were determined by enzymatic hydrolysis [15], and total carbohydrates were calculated by difference. The pH and total titratable acidity (TTA) were determined using the method proposed by Robert et al. [17].

### 2.4. Bread Characterization

The pieces of bread (crust and crumb) were evaluated on days 1, 3, 5, and 7 after baking by crumb moisture analysis using method Nº 44-15.02 [18]. The water activity (aw) of the crumb was determined using an AquaLab digital water activity meter model PRE (Decagon, Pullman, DC, USA), specific volume was calculated by dividing the volume of the bread (cm^3^) by its mass (g), according to method 10-05.01 of AACC [18], and firmness was evaluated according to method 74.09 using a texturometer TA.XT plus (Stable Micro System, Godalming, UK) [18]. Color quantification was performed using a Color Quest II Hunterlab spectrophotometer (HunterLab, Reston, VA, USA) in transmittance mode according to the CIE-Lab color system. L or luminosity values [(zero (dark)/100 (light)], and chromaticity, defined by a (+a = red and −a = green) and b (+b = yellow and −b = blue), were determined. With these parameters, the cylindrical coordinates C (chroma) and h (hue angle), which define the intensity and tone of the samples, and the color differences (ΔE) between the samples and a control bread were calculated according to Equation (1) [19], Equation (2) [19], and Equation (3) [20], respectively.
(1)$$C=\sqrt{a^{2}+b^{2}}$$
(2)$$h=\tan g\;(\frac{a}{b})$$
(3)$$∆E=\sqrt{({∆L)}^{2}+({∆a)}^{2}+{\left(∆b\right)}^{2}}$$
where ΔE means the difference between the JPF bread and the control bread.

### 2.5. Antioxidant Analysis

A hydroethanolic extract was prepared to characterize the total reducing capacity and antioxidant capacity using the ultrasound method (Sonics Vibra-Cell VCX750, Newtown, CT, USA) and a Sonics microprobe of 13 mm in diameter and a frequency of 720 Khz for 3 min, following the methodology used by Tarone et al. [21]. The contact height of the probe with the extracts was standardized to 40 mm. An ice bath was used to prevent overheating of the extracts, with a maximum temperature of 40 °C measured at the system. In each experiment, 3 g of JPF or bread was mixed with 30 mL of 50% ethanol (*v*/*v*). After ultrasound processing, the samples were centrifuged at 11,180× *g* force for 10 min, and the supernatant was collected in a single extraction. All analyses were performed in triplicate.

The total reducing capacity was performed using the Folin-Ciocalteau method described by Swain and Hillis [22], using a standard curve prepared with gallic acid (20, 40, 60, 80, and 100 mg/L). A 50 µL aliquot of the extracts was added to 800 µL of water and 50 µL of Folin-Ciocalteau reagent for the reaction. The mixture was incubated for 2 h in the dark, then 100 µL of 7.5% sodium carbonate (*w*/*v*) was added, and the absorbance was read at 725 nm. The results were expressed as mg gallic acid equivalents (GAE) per gram.

The oxygen radical absorbance capacity (ORAC) measure was used to determine the ability of the extract to scavenge free radicals [23]. For the assay, 25 µL of extract, 150 µL of fluorescein, and 25 µL of AAPH (2,2′-azobis (2-methylpropionamidine)) were added to a black microplate. The AAPH solution was prepared just before being placed on the microplate. The extract was replaced with potassium phosphate buffer (pH 7.4) for the blank assay. The fluorescence was read using emission filters at 520 nm and excitation at 485 nm every 1 min for 120 min. The results were expressed in µmol Trolox equivalents (TE) per gram.

The HPLC method for anthocyanin determination was adapted from Faria, Marques, and Mercadante [24]. The analyses were conducted on an HPLC (Agilent Technologies, 1200 series, Santa Clara, CA, USA) system equipped with a diode array detector. The anthocyanins were separated with a C18 column (Zorbax, 5 µ, 4.6 mm × 250 mm, Agilent Technologies, Santa Clara, CA, USA) by a linear gradient of methanol:water, both with 5% formic acid, from 90:10 to 60:40 in 20 min and 20:80 in 15 min, maintaining this proportion for 5 min. Chromatograms were processed at 520 nm. The identification was based on elution order on the C18 column, relative retention time, and spectral data compared to standards analyzed under the same conditions. The quantification was carried out using an analytical curve built with cyanidin 3-glucoside.

### 2.6. Microbiological Analysis

The microbiological analysis of the bread followed the recommendation of the Brazilian National Health Surveillance Agency of the Ministry of Health (ANVISA), based on Resolution Nº 12 [25], with analysis being carried out to verify the presence of coliforms and salmonella in samples of the four bread formulations, using the filter membrane method [26]. Viable lactic acid bacteria (LABs) were evaluated in dough and bread samples (crust and crumb) during all production stages and at 1-day intervals until spoilage. First, a 25 g sample of bread was diluted in 225 mL of 0.1% peptone water (bacteriological peptone, *w*/*v*), and then a serial dilution was made from 10^−1^ to 10^−8^. One milliliter of each dilution was transferred to the plates and added to the specific medium, MRS agar (DIFCO (Beirut, Lebanon), Becton Dickinson (Franklin Lakes, NJ, USA), Sigma-Aldrich Brasil Ltda, Barueri, SP, Brazil). The plates were incubated for 48 h at 37 °C, and the colonies were counted using the colony counter (ZEISS-17p). Molds and yeasts (total fungi) were also evaluated in the samples after five days of incubation at 25 °C; 1 mL of each serial decimal dilution was seeded in depth, in duplicate, using potato dextrose agar (BDA) acidified with 10% tartaric acid to pH 3.5.

### 2.7. Clinical Trial

The clinical trial was approved by the University Research Ethics Committee (CAAE:06800619.9.0000.5404) before it started, and the guidelines of Resolution 466 of the National Health Council for research with human beings were respected [27], as well as the Declaration of Helsinki.

#### 2.7.1. Recruitment of Individuals

Twelve healthy individuals (BMI 18.2–24.9 kg/m^2^) aged 21–38 were recruited for the single-blind, placebo-controlled, randomized, and crossover clinical trial. The inclusion criteria were not being a smoker, not having any known metabolic disorder or food allergy, absence of chronic diseases, and not using medications, nutritional supplements, antibiotics, and/or probiotics in the 30 days before the start of the trial, or even during the experimental period.

#### 2.7.2. Experimental Protocol

The tests were performed in two stages with one-week intervals between evaluations, and all participants read and signed the Free and Informed Commitment Term before starting the experiments.

The JPF2 formulation was selected for this clinical trial based on its chemical and microbiological results. The bread used in the experiments was prepared 18 h before the test day, and the portion offered to each individual was defined to provide 50 g of available carbohydrates (112 g of JPF2 bread and 128 g of control bread); the available carbohydrate was analyzed according to McCleary et al. [28].

The individuals were instructed to fast for 12 h on the morning of the test. Upon arriving at the clinical laboratory, the individuals were placed in armchairs and rested for 15 min before starting the test. An intravenous catheter was inserted into the antecubital vein of each participant, and a sample of venous blood (10 mL) was collected (basal period, time 0). All received a test meal consisting of a slice of bread (JPF2 or control) and a 250 mL glass of water to be consumed within 10 min. After consuming the meal, blood samples were collected in the postprandial period at times 15, 30, 45, 60, 90, 120, and 180 min to evaluate the glycemic response using the YSI 2700D biochemical analyzer (Marshall Scientific, Hampton, NH, USA), insulinemia by chemiluminescence (performed in an outsourced clinical analysis laboratory), and serum antioxidant capacity by the previously described ORAC method. Anthropometric parameters, including weight (kg), height (cm), and waist and arm circumferences (cm) were measured. Individuals’ glycemic and insulinemic responses to JPF2 bread were calculated based on the area under the curve and the incremental area compared to the results obtained by the control bread.

The insulin resistance index (HOMA-IR) and the insulinogenic index (IGI) were used to measure β-cell function. These indices are based on statistical models using stepwise linear regression analysis. The variables in the model assumed the availability of determinations at 0, 30, 60, 90, and 120 min. The HOMA-IR index was calculated using the following formula: HOMA-IR = ((insulin fasting mU/L) × (glucose fasting mg/dL)/405). IGI was calculated with the following formulas: IGI = (insulin30 min–insulin fasting/glucose 30 min–glucose fasting) using the model https://mmatsuda.diabetes-smc.jp/MIndex.html (accessed on 17 January 2024) [29].

The subjective appetite profile concerning the consumption of JPF bread was assessed using a visual analog scale (VAS) in which individuals were questioned regarding current feelings of hunger, satiety, fullness, and desire for food, anchored by the terms “not at all” and “extremely” throughout the blood collection times (0, 30, 45, 60, 90, 120, and 180 min) after the bread intake [30].

### 2.8. Statistical Analysis

The physicochemical analysis results were performed using univariate ANOVA statistics, and the means were compared using Tukey’s test at 5% significance. The statistical program used was GraphPad Prism 5.0 (GraphPad Software, Inc., La Jolla, CA, USA). The Kolmogorov–Smirnov test was applied to assess data normality. Parameters with normally distributed data were analyzed using the paired *t*-test and presented as mean values and standard deviations. The non-normal variables insulin and ORAC (mmol TE/mL) were transformed before the parametric test and presented as least squares mean and standard deviations. The incremental peak and AUC (area over the curve)were calculated for the control and test for each participant using the trapezoidal model [31].

For individual data from the clinical trial and the area of the total glycemic response curve, the results were analyzed using two-way ANOVA statistics, followed by Bonferroni tests and the paired Student’s *t* test for the area under the curve, and the means were compared by Tukey’s test at 5% significance, using GraphPad Prism 5.0 software.

## 3. Results and Discussion

### 3.1. Physicochemical Evaluation and Technological Parameters of the Bread

The bread developed in the experiment was based on the classic artisanal bakery technique using acid and slow fermentation with adapted type II sourdough. The approximate composition of jabuticaba peel flour (JPF) was previously analyzed and described by Loubet Filho et al. [32], with approximately 34% of dietary fiber observed, of which 74% represented the insoluble fraction. Table 2 presents the approximate composition of the JPF and control bread formulations. The data showed that adding JPF to the formulations modified the nutrient content of the matrices. The increase in protein, lipids, ash, total fiber, and insoluble fiber in the JPF2 and JPF3 formulations occurred due to the inclusion of JPF in the formulation; therefore, this supplementation increased the nutrient supply and nutritional composition of the final product, with a significant increase of more than 50% in total and insoluble fibers and ash (minerals).

Bread with JPF had a significantly lower pH (≤4.7) compared to the control bread (pH = 5.4) (*p* < 0.05) (Table 3), which may be due to the presence of lactic acid bacteria (LABs) capable of producing different types of bioactive molecules, such as organic acids, fatty acids, hydrogen peroxide, and bacteriocins [33]. In addition, during the fermentation process, phenolic compounds may have been released from the matrix of the JPF, thus modifying the pH of the final product; these values are similar to those found with the addition of fruit, such as pitaya pulp (pH = 4.3) and Andean blueberry (pH = 4.7), to long-fermentation bread [34,35].

Gänzle [7] described the mechanism governing the decrease in pH during lactic acid fermentation, with the action of intracellular peptidases from lactic acid bacteria being responsible for the accumulation of thiols that break the disulfide bonds of the gluten protein, increasing its solubility. If the pH of the dough is below 4.5, the maltogenic amylases in the fresh dough are inhibited; however, dextrin and maltodextrin are still released by glucoamylase.

The JPF2 formulation showed the highest firmness (force/g), followed by JPF3 and JPF1, and the control bread remained at the lowest values throughout the storage period, showing greater softness. JPF is a source of fiber, which may have contributed to increased firmness. Firmness had a progressive increase in all formulations during the shelf life, which was more significant in bread with JPF, since mold development was observed in the control bread from the fifth day of storage (Table 3); an increase in firmness during the shelf-life period is expected in bakery products [36]. Therefore, the results suggest that adding JPF to the formulation also acted as a natural preservative in the bread due to the presence of polyphenols [34] that protected against mold formation.

Regarding color evaluation, luminosity differed among the formulations due to the addition of JPF (Table 3). The red/green coordinate (a) was lower only for the control. It increased in the formulations with the addition of JPF, as did the concentration of purple pigment (dark color) and the yellow/blue coordinates (b) for JPF3. The increased concentration of JPF in the formulation gave the bread a different color, varying with the intensity of the purplish pigment, and, empirically, increased the aroma, making the bread attractive to the consumer (Figure 2).

The specific volume of the formulations was above 3.70 cm^3^/g, ranging from 3.80–3.98 cm^3^/g, with no statistical differences, although the control bread presented a higher specific volume (Table 3).

A similar specific volume was observed in breads that had whey added to their formulation for enrichment (4.0 cm^3^/g) by Lima et al. [37], and values above 3.70 cm^3^/g with the use of the additive ascorbic acid [38]. Replacing the calcium propionate additive and flour improver with JPF provided adequate bread volume and a similar growth and fermentation profile.

Adding JPF to the dough decreased the total fermentation time by 30–40 min compared to the control sample, which was 820 min, observed when the dough reached twice its initial volume. This result was expected due to the composition of JPF, which provides fermentable carbohydrates [39] for native Saccharomyces yeasts, yeasts added to the mixture, and heterofermentative bacteria in the medium [40].

### 3.2. Bread Stability (Shelf Life)

The stability or shelf life of bakery products is achieved by controlling the growth of molds and yeasts, which is influenced mainly by environmental conditions (storage temperature, oxygen concentration in the packaging, and humidity) [41]. Commercial preservatives or some natural equivalents are usually used to extend this product’s shelf life. Considering that JPF has a considerable quantity of phenolic compounds in its composition, some with antimicrobial activity, the assumption exists that its addition to the formulation could somehow act as a natural preservative agent [13,14].

We emphasize that no commercial additives were added to the formulations in the present study. One of the reasons for not adding them was to produce an artisanal “clean label” bread; another was to evaluate the impact on bread preservation of adding JPF. Although high water activity was observed in all formulations (Table 3), the control bread samples presented pH values higher than 4.5, favoring microbial growth (Table 4) after five days of baking, which is expected in long-fermentation bread [42]. On the other hand, the formulations containing JPF showed a pH lower than 4.7, giving them a preservative effect and ensuring greater viability for consumption up to the seventh day of shelf life. Lower pH improves the metabolism of lactic acid bacteria and the production of postbiotic metabolites that increase the bread’s shelf life [43].

The extension of shelf life to more than seven days has been reported in studies with such added ingredients as grains and fruits [44,45,46,47]. Some studies have demonstrated that heterofermentative lactic acid bacteria, such as those present in JPF, can act as preservatives in baked goods due to the production of propionic acid during fermentation [48,49,50] that is inhibitory against certain microorganisms, such as *Pseudomonas aeruginse*, *Listeria monocytogenes*, and *Staphylococcus aureus* [51,52]. Furthermore, the results of the microbiological analysis demonstrate that the breads comply with current Brazilian legislation and are safe for human consumption.

### 3.3. Stability of Antioxidant Compounds after the Fermentation and Baking Processes

The antioxidant capacity and total reducing capacity of the control and JPF breads are shown in Table 5. The increment in these parameters in JPF bread could be associated with the autolysis of the flour during fermentation, which contributes to the release of bioactive peptides and phenolic compounds associated with the wheat flour and JPF matrix; these have antioxidant capacity and/or reactivity to the Folin-Ciocaulteau reagent. The fermentation and baking processes could also contribute to increasing these parameters due to the products of the Maillard reaction, such as melanoidins and acrylamide, in the bread crust [53]. Melanoidins are heterogeneous polymers with high molecular weight, generated in the last stages of the Maillard reaction and generally found in greater quantities the shorter the fermentation time [54].

The orange tree honey added to the formulation (reduced sugars) also reacts with the Folin-Ciocaulteau reagent; however, the highest concentration of honey was added to the control formulation, presenting the lowest antioxidant capacity and reducing capacity values. In this sense, the increase in antioxidant capacity and reducing capacity observed in formulations containing JPF must be related to the compounds present in the fruit peel, which can be demonstrated by the increase in the percentage of JPF added to the formulation. Adding JPF to the formulations provided the bread with anthocyanins from the jabuticaba peel that were not present in the control formulation.

Polyphenol stability was one of the concerns about adding JPF to a product subjected to baking conditions with temperatures up to 188 °C. However, when comparing the different formulations, a positive correlation (r = 0.8) between the increase in JPF concentration in the formulation and the increase in antioxidant capacity and total reducing capacity was noticed, indicating that even with the degradation of some compounds, there would be no loss of the desired properties in the final product.

We can see in Table 5 that the increase in antioxidant capacity and total reducing capacity was more significant from JPF1 to JPF2 than from JPF2 to JPF3. The increase in antioxidant capacity and reducing capacity from the JPF1 formulation to JFP2 agrees with expectations, since it is related to the proportional increase in the concentration of JPF in the formulation. However, the same increase was not observed when we compared the JPF2 formulation to the JFP3 formulation. This behavior could be explained by the issue of saturation of the reaction medium, leading to a smaller number of reaction sites and a lower response. Therefore, in addition to evaluating the samples’ color, aroma, and flavor, we selected the JPF2 formulation for the clinical trial. The jabuticaba peel contains many tannins in its composition, adding a characteristic and intense fruity flavor to the bread. Furthermore, anthocyanins change the bread color (purple appearance), making the bread exotic to consumers’ habits.

We can also observe in the chromatographic analysis a considerable difference in the quantification of anthocyanins (2.49 times) in the JPF3 samples compared to the JPF1 sample; however, the difference between the JPF2 and JPF1 samples was 1.13 times (Table 6). Considering that the increase in antioxidant activity in sample JPF3 was only 16% compared to sample JPF2, and considering the sensory aspects, we selected the sample with 7.5% JPF for the clinical trial.

### 3.4. Clinical Trial

The nutritional characteristics of the 12 individuals recruited for the clinical trial are shown in Table 7. In Figure 3A, we observe the glycemic response curve of individuals when eating the control bread and the bread with added JPF. In the control bread, the glycemia peak occurred 30 min after consumption and remained high until 45 min, when gentle inflection began. Since it is a bread produced using the long-fermentation method, the drop in blood glucose is not as steep as we generally see in traditional white bread [55,56]. For bread with JPF, we have a different behavior in the curve, where a less prominent peak at 45 min and a decrease with a smaller inflection until 180 min occurred, resulting in a smaller area under the curve, as shown in Figure 3A.

The increase in blood glucose that characterizes the glycemic peak observed in Figure 3A occurs due to glucose absorption after food ingestion (postprandial stage). In an unfed state, we maintain our blood glucose at the basal level necessary to maintain vital functions. However, when we eat foods especially rich in carbohydrates, the digestion process releases monosaccharide molecules into the intestine. These include glucose, which is absorbed and released into our circulation [57].

The increase in glucose levels in circulation in the postprandial stage signals our pancreas to secrete the counterregulatory hormone insulin. Insulin, in turn, signals to insulin-dependent tissues that there are large amounts of glucose in the circulation, which must be stored to return blood glucose levels to the basal level. In this way, the control of glycemia is partly due to the characteristics of the food we eat, the digestion and absorption capacity of our body, and the release and effective signaling of insulin in the tissues. Failures in the insulin signaling process can lead the individual to develop insulin resistance, a clinical condition also known as a prediabetic state.

In Figure 3B,D, no significant difference in insulin response (*p* > 0.05) between the control and JPF bread can be observed. However, the postprandial blood glucose level spent more time returning to a basal level for the bread with added JPF than for the control bread (*p* < 0.05). This observation agrees with the greater incremental area under the curve (AIUC, Figure 3D) observed in the JPF bread group compared to the control bread group. The differences observed in the glycemic curve and the insulin response may be associated with the supplemental amount of phenolic compounds, as shown in Table 6, in which cyanidin-3-glucoside was identified as the main compound and in greater quantity, and with the fibers added to the bread from the jabuticaba peel [58].

There are reports in the literature that some phenolic compounds can inhibit digestive enzymes, such as glycosidases [59,60]. In this way, adding JPF to the bread formulation improves its nutritional functionality because the changes in the glycemic and insulinemic responses of individuals demonstrate greater control of glycemia and can benefit the health of those who consume it.

In this sense, long-fermentation bread products, such as sourdough or levain-based bread, have moderate GI values of between 56 and 69 [61,62]. When they are made from ingredients of whole grains or contain fiber, there is an even greater reduction in the index to slow down the release and absorption of glucose in the gastrointestinal tract [63].

In long-fermentation bread, in which a slow fermentation process develops, the dough microbiota can partially or entirely degrade carbohydrates, free sugars, and proteins. Once used as a substrate by lactic acid bacteria and yeast to form the fermented mass, these nutrients reduce the carbohydrates available in the final product, which are converted into organic acids, volatile compounds, and peptides of nutritional interest [64].

In addition to the results observed in the glycemic and insulinemic response analyses, the serum oxygen radical absorption capacity was monitored for 180 min after the control and JPF bread were consumed. Figure 4 shows a considerable increase in the serum antioxidant capacity of the individuals when they ingested samples containing JPF. Even the control samples showed increased antioxidant capacity, which may be related to the intestinal absorption of bioactive peptides formed in the long-term fermentation process. However, in bread containing JPF, an increase was observed as well as the maintenance of antioxidant capacity for a more extended period. This result may be associated with the absorption of bioactive compounds in JPF and the secondary products generated during bread fermentation. Furthermore, this result may corroborate previous data, indicating that there is indeed an increase in serum bioactive compounds, and that these may be related to the results of the glycemic and insulinemic response.

This means that the glycemic curve was maintained in the presence of lower absolute insulin production, corresponding to better insulin action, potentially amplified with some incretin effect (increased GLP-1) and reduced inflammation (Geraldi et al., 2021 [65]); this resulted in improved insulin sensitivity, as shown in Table 7, with a significant reduction in the IGI score from 112 to 88 (*p* = 0.01).

Geraldi et al. [65] observed similar results in healthy individuals aged 18–40 years after ingesting jabuticaba juice. Both studies observed an increment in and the maintenance of antioxidant capacity for more than 2 h after jabuticaba fruit intake. Similarly, Plaza et al. [66] observed an increase in the antioxidant capacity of healthy individuals (26.5 ± 3.4 years) after consuming JPF, which was maintained for up to 6 h and influenced the glycemic and insulinemic response of a second meal eaten (after 3 h). In this way, consuming bread containing JPF can significantly promote health for individuals and provide an optional alternative to traditional white bread.

Additionally, the glycemic index of carbohydrates has been associated with promoting satiety because the return of blood glucose to basal levels is slower than that of carbohydrates with a high glycemic index. In their bibliographical review, Bornet et al. [67] gathered 19 studies, including 248 individuals, which proved that consuming foods with a low glycemic index promoted a greater feeling of satiety than foods with a high glycemic index.

In agreement with the data obtained in the glycemic and insulin curves, the feeling of hunger reported by the individuals during the 180 min of the test was lower in the group that consumed the JPF bread; however, no statistically significant difference was observed between the groups. Likewise, the desire to eat other foods was also lower in the JPF bread group, especially after 60 min (no statistically significant difference was observed between the groups, Appendix A).

Although no statistically significant difference was observed between the experimental groups regarding the sensation of satiety and fullness after 60 min, the reports from the JPF group seemed to be greater than those from the control group. We highlight that the VAS is a subjective assessment; psychological, environmental, and product preference factors can interfere with the individual’s perception, and the number of test participants was small. In any case, the results observed here corroborate previous data and reinforce that the consumption of bread containing JPF can contribute to the health of its consumers as a nutritious and functional option.

## 4. Conclusions

The long-fermentation process in bakery products substantially improves the nutritional quality of the bread. It also favors changes in the bioaccessibility and bioavailability of nutrients in the food matrix. Furthermore, including JPF in the bread formulation provided a longer shelf life.

Bread produced using the sourdough method has a lower glycemic response. Adding JPF promoted an even greater decrease in the glycemic response, reducing the postprandial glucose peak and promoting the maintenance of glycemia for a more extended period. An improvement in insulin response was also observed, suggesting that consuming bread containing JPF could be an alternative for individuals needing to control blood glucose and adopt a more restrictive diet.

## Figures and Tables

**Figure 1 foods-13-02878-f001:**
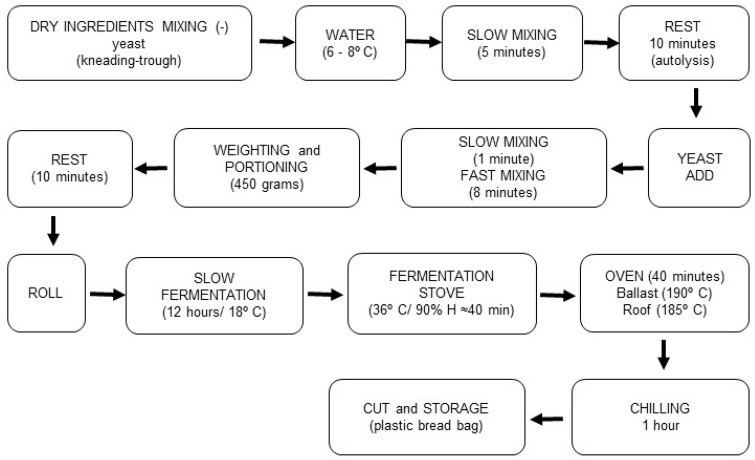
Flowchart for preparing and manufacturing slow fermentation artisanal bread—adapted type II sourdough method.

**Figure 2 foods-13-02878-f002:**
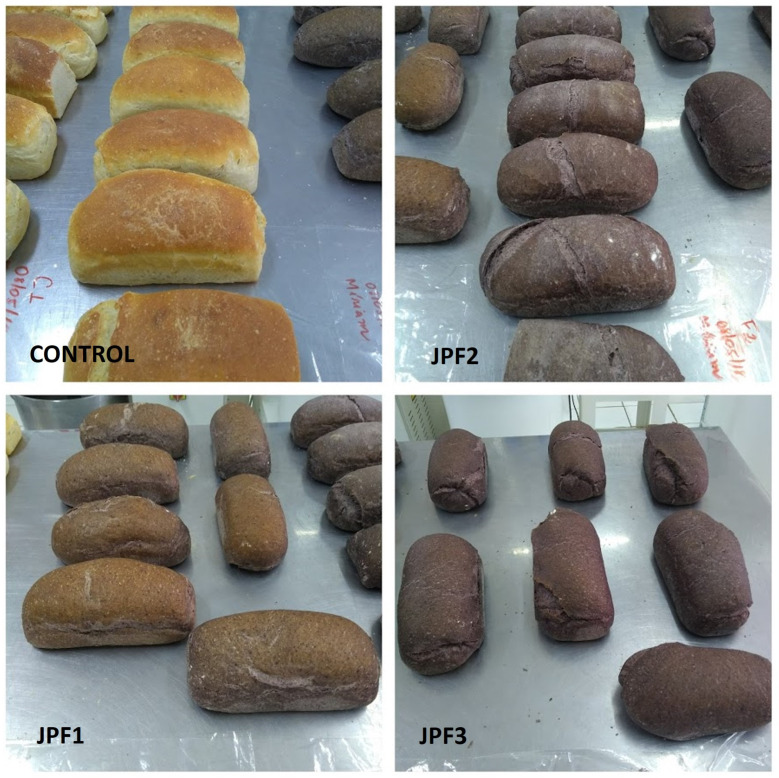
A representative image of bread produced without and with the addition of jabuticaba peel flour (JPF1 = 5%; JPF2 = 7.5%; JPF3 = 10%).

**Figure 3 foods-13-02878-f003:**
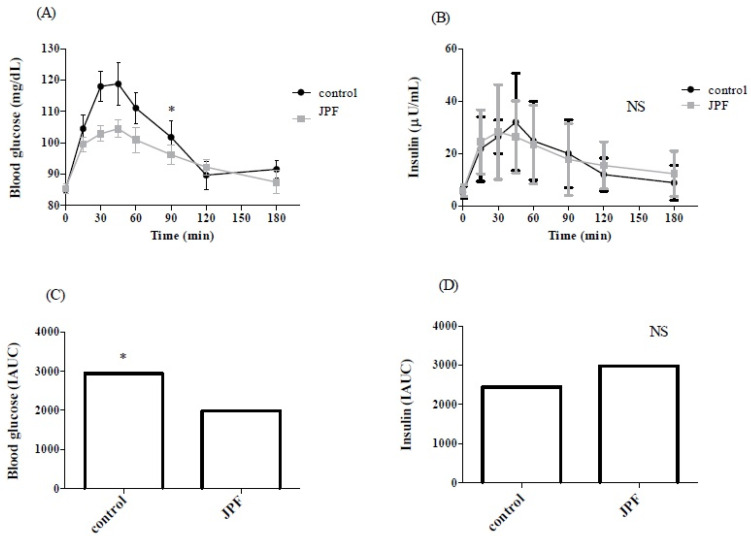
Glycemic (**A**) and insulinemic (**B**) responses after consumption of bread with jabuticaba peel flour (JBF) or control by healthy individuals. NS = not significant; IAUC = incremental area over the curve. * Indicates statistical differences when analyzed by two-way ANOVA and Bonferroni post-test (**A**,**B**) and Student’s *t* test (**C**,**D**), considering *p* < 0.05.

**Figure 4 foods-13-02878-f004:**
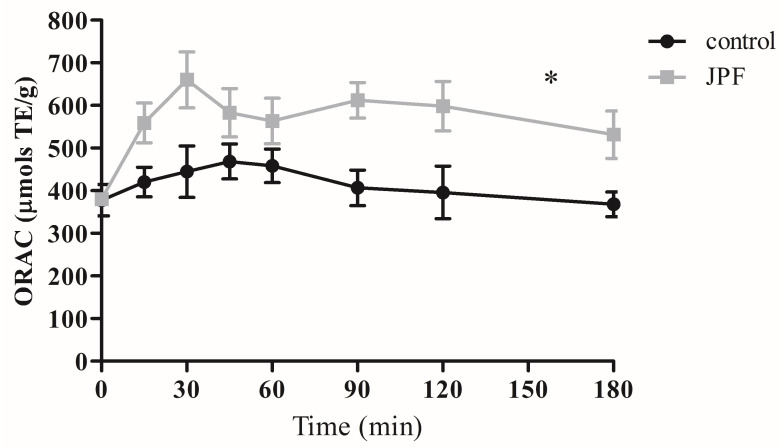
Oxygen radical absorbance capacity (ORAC) postprandial concentration in healthy individuals after consumption of JPF bread. * Indicates statistical differences when analyzed by two-way ANOVA and Bonferroni post-test, considering *p* < 0.05.

**Table 1 foods-13-02878-t001:** Formulation of slow fermentation artisanal bread—an adaptation of the type II sourdough method, including jabuticaba peel flour (JPF).

Ingredients	Bread Formulation (*)			
	Control	(JPF1)	(JPF2)	(JPF3)
Wheat flour	100.0	95.0	92.5	90.0
JPF	-	5.0	7.5	10.0
Water	69	78	78	76
Orange tree honey	4.00	1.17	0.65	-
Biological yeast	2	2	2	2
Salt	1.8	1.8	1.8	1.8

(*) Approximate % of the weight of wheat flour on a wet basis.

**Table 2 foods-13-02878-t002:** Approximate composition (g/100 g) of sourdough bread with added jabuticaba peel flour (JPF).

	Formulation			
Parameter	Control	JPF1	JPF2	JPF3
Protein	9.9 ± 0.18 ^b^	10.0 ± 0.001 ^b^	10.3 ± 0.26 ^a^	10.4 ± 0.09 ^a^
Lipids	1.7 ± 0.07 ^b^	1.7 ± 0.05 ^b^	1.8 ± 0.24 ^a^	1.8 ± 0.27 ^a^
Ash	1.1 ± 0.04 ^c^	2.3 ± 0.02 ^b^	2.3 ± 0.01 ^b^	2.5 ± 0.01 ^a^
Total fiber	1.0 ± 0.04 ^c^	1.0 ± 0.02 ^c^	2.3 ± 0.11 ^b^	2.9 ± 0.07 ^a^
Insoluble fiber	0.8 ± 0.03 ^c^	1.0 ± 0.002 ^b^	1.0 ± 0.003 ^b^	2.0 ± 0.03 ^a^
Total carbohydrates	42.47 ^a^	40.64 ^b^	40.83 ^b^	40.88 ^b^

Different lowercase letters in the same line represents a statistical difference between the samples (*p* < 0.05).

**Table 3 foods-13-02878-t003:** Physicochemical and physical–technological parameters of control bread and bread with added jabuticaba peel flour (JPF) during their shelf lives.

		Formulations			
Parameters	Days	Control	JPF1	JPF2	JPF3
pH	1	5.4 ± 0.02 ^d^	4.7 ± 0.03 ^c^	4.5 ± 0.02 ^b^	4.4 ± 0.01 ^a^
	3	5.3 ± 0.01 ^d^	4.7 ± 0.01 ^c^	4.5 ± 0.01 ^b^	4.6 ± 0.01 ^a^
	5	5.3 ± 0.02 ^d^	4.7 ± 0.09 ^c^	4.5 ± 0.03 ^b^	4.6 ± 0.01 ^a^
	7	n.d.	4.6 ± 0.08 ^c^	4.4 ± 0.01 ^a^	4.5 ± 0.01 ^b^
Moisture	1	44.6 ± 0.40 ^aA^	45.0 ± 0.25 ^aA^	44.8 ± 0.20 ^aA^	43.6 ± 0.55 ^aA^
	3	43.5 ± 0.90 ^aA^	44.2 ± 0.76 ^aA^	43.9 ± 1.29 ^aA^	40.7 ± 2.03 ^bB^
	5	42.1 ± 0.70 ^bA^	41.3 ± 0.80 ^bA^	42.6 ± 0.65 ^bA^	38.9 ± 0.71 ^cB^
	7	40.7 ± 0.31 ^cA^	39.3 ± 1.19 ^cA^	39.0 ±1.30 ^cA^	37.7 ± 0.55 ^cB^
Water activity (aw)	1	0.94 ± 0.02	0.94 ± 0.01	0.95 ± 0.00	0.95 ± 0.00
	3	0.90 ± 0.00	0.90 ± 0.00	0.90 ± 0.00	0.90 ± 0.00
	5	0.95 ± 0.01	0.96 ± 0.00	0.96 ± 0.01	0.95 ± 0.01
	7	0.90 ± 0.01	0.90 ± 0.00	0.90 ± 0.00	0.90 ± 0.01
Texture (force/g)	1	1083.12 ^a^	1363.43 ^b^	1863.11 ^d^	1797.79 ^c^
	3	2042.10 ^a^	2194.12 ^b^	2220.65 ^c^	2458.54 ^d^
	5	2213.05 ^a^	2273.96 ^a^	2737.37 ^b^	2683.68 ^b^
	7	n.d.	2876.35 ^a^	2785.61 ^a^	3106.27 ^b^
Color	L	78.7 ± 0.31 ^a^	37.7 ± 1.27 ^b^	31.8 ± 0.32 ^c^	28.3 ± 0.47 ^d^
	a	5.2 ± 0.10 ^a^	44.8 ± 0.61 ^b^	50.9 ± 0.50 ^bc^	58.6 ± 0.32 ^c^
	b	16.9 ± 0.30 ^a^	5.4 ± 0.58 ^b^	4.7 ± 0.41 ^bc^	4.5 ± 0.15 ^c^
	C	17.23	25.80	45.11	63.33
	ΔE	3.9	27.9	46.6	64.7
Specific volume (cm^3^/g)		4.0 ± 0.06 ^a^	3.8 ± 0.08 ^b^	3.8 ± 0.05 ^b^	3.8 ± 0.08 ^b^

n.d.= not determined (bread developed mold). pH, aw, texture, color, specific volume = different lowercase letters in the same row represent statistical differences between samples (*p* < 0.05). Moisture = different capital letters in the same row or different lowercase letters in the columns represent statistical differences between samples (*p* < 0.05).

**Table 4 foods-13-02878-t004:** Evaluation of microbiological growth during the determination of the shelf life of control bread and bread with added jabuticaba peel flour (JPF).

		Formulations			
Parameters	Days	Control	JPF1	JPF2	JPF3
Molds (log CFU/g)	1	<1 *	<1	<1	<1
	3	<1	<1	<1	<1
	5	2.77 ± 0.60 ^b^	<1	<1	<1
	7	4.80 ± 0.04 ^a^	<1	<1	<1
Fungi (log CFU/g)	1	<1	<1	<1	<1
	3	<1	<1	<1	<1
	5	4.40 ± 0.07 ^b^	<1	<1	<1
	7	5.47 ± 0.03 ^a^	<1	<1	<1
Lactic acid bacteria (log CFU/g)	1	<1	2.21 ± 0.1 ^c^	5.99 ± 0.29 ^c^	1.68 ± 0.33 ^d^
	3	6.03 ± 0.30 ^a^	5.08 ± 0.20 ^b^	9.08 ± 0.14 ^b^	5.16 ± 0.37 ^a^
	5	3.74 ± 0.41 ^c^	4.97 ± 0.09 ^b^	9.18 ± 0.08 ^b^	3.89 ± 0.36 ^c^
	7	5.89 ± 0.56 ^b^	6.46 ± 0.34 ^a^	10.23 ± 0.72 ^a^	4.05 ± 0.07 ^b^
Yeasts (log CFU/g)	1	<1 **	<1	<1	<1
	3	<1	<1	<1	<1
	5	2.89 ± 0.26 ^b^	<1	<1	<1
	7	6.93 ± 0.18 ^a^	5.48 ± 0.78 ^a^	1.88 ± 0.69 ^a^	2.34 ± 0.24 ^a^

* detection limit < 15 CFU; ** < 25 CFU. Different lowercase letters in the lines represent statistical differences between samples (*p* < 0.05).

**Table 5 foods-13-02878-t005:** Antioxidant capacity of phenolic compounds after the bread baking process.

Formulation	ORAC (µmol TE/g)	IAA (x)	TRC (mg GAE/g)	ITRC (x)
Control bread	123.0 ± 6.241 ^c^	-	64.1 ± 3.997 ^d^	-
JPF1 bread	166.6 ± 4.172 ^c^	1.35	99.9 ± 3.839 ^c^	1.56
JPF2 bread	373.1 ± 2.960 ^b^	3.03	144.9 ± 1.224 ^b^	2.26
JPF3 bread	434.5 ± 8.973 ^a^	3.53	171.8 ± 1.894 ^a^	2.67
JPF	929.8 ± 1.201	-	115.8 ± 0.825	-

ORAC = oxygen radical absorbance capacity. IAA = increase in antioxidant activity compared to control bread. TRC = total reducing capacity (total phenolics). ITRC = increase in total reducing capacity compared to control bread. Different means with lowercase letters in the columns differ significantly from each other according to Tukey’s test (*p* < 0.05).

**Table 6 foods-13-02878-t006:** Anthocyanin content after baking the bread (mg/100 g of bread).

Formulation	Anthocyanin Content
Control bread	NS *
JPF1 bread	0.040 ± 0.0002 ^c^
JPF2 bread	0.045 ± 0.0005 ^b^
JPF3 bread	0.099 ± 0.0009 ^a^

* NS = not significant. Different means with lowercase letters in the columns differ significantly from each other according to Tukey’s test (*p* < 0.05).

**Table 7 foods-13-02878-t007:** Nutritional status of clinical trial participants.

Parameter	Men (*n* = 4)	Women (*n* = 8)
	Mean ± SD	
Age (years)	30.5 ± 4.04	29.9 ± 5.08
Weight (kg)	73.2 ± 3.09	62.3 ± 8.22
BMI (kg/m^2^)	24.3 ± 0.54	22.7 ± 1.52
Neck circumference	27.5 ± 0.50	29.8 ± 3.25
Abdominal circumference	80.7 ± 8.08	76.5 ± 9.96
Arm circumference	33.0 ± 1.00	28.5 ± 4.77
HOMA-IR	1.2 ± 0.88	1.3 ± 0.80
IGIest (μmol/kg per min per pmol) *		
Control bread	112.2 ± 14.97	
JPF bread	88.0 ± 22.84	

HOMA-IR = insulin resistance index; IGI = insulinogenic index. * Values are least-squares means (LSM) and standard deviations.

## Data Availability

The original contributions presented in the study are included in the article, further inquiries can be directed to the corresponding author.

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
