# Peer review of "Production of Long Fermentation Bread with Jabuticaba Peel Flour Added: Technological and Functional Aspects and Impact on Glycemic and Insulinemic Responses"

_foods, 2024, doi:10.3390/foods13182878_

Round 1
Reviewer 1 Report
Comments and Suggestions for Authors
1. The author's affiliation is consistent, and the author only needs to write one affiliation. You can refer to the affiliation signature of the published article.
2. There are too many introductory paragraphs. It is recommended to only introduce the beneficial effects of additives such as bread and jabuticaba on bread, and then introduce the significance of this study in the last sentence.
3. The author should conduct a thorough analysis of the impact of added jabuticaba peel flower on bread quality, especially the possible reasons for the influence of different amounts and physicochemical properties of added jabuticaba peel flower on bread quality?
4. Is the effect of Jabitica peel bloom during shelf life on the shelf life of packaging based on the influence of its own system pH on microorganisms or the generation of other functional components to exert antibacterial effects.
5. The image quality or pixel count of the entire text is slightly low. It is recommended that the author provide high-quality images in the text. Suggest adding a picture of bread in the image to help reviewers or authors identify differences.
6. In the conclusion, the author needs to elaborate on which beneficial qualities of bread can be improved by adding jabuticaba peel flower to bread.
Comments on the Quality of English LanguageNone
Author Response
Comments 1: The author's affiliation is consistent, and the author only needs to write one affiliation. You can refer to the affiliation signature of the published article.
Response 1: We have changed the description of the authors' affiliation.
Comments 2: There are too many introductory paragraphs. It is recommended to only introduce the beneficial effects of additives such as bread and jabuticaba on bread, and then introduce the significance of this study in the last sentence.
Response 2: We agree with the reviewer that our introduction was more extended than desired. Still, it is important to contextualize our choices in detail to make it easier for the reader to understand the proposal of the manuscript. We went into a little more detail about the production process of long-fermentation bread and the importance of valuing the jabuticaba peel to justify the steps and results of the work. Since the other three reviewers did not question the introductory text, we decided it would be better to keep it as it is.
Comments 3: The author should conduct a thorough analysis of the impact of added jabuticaba peel flower on bread quality, especially the possible reasons for the influence of different amounts and physicochemical properties of added jabuticaba peel flower on bread quality?
Response 3: We could not understand what the reviewer requested since the paper presents all the physicochemical and technological modifications related to including jabuticaba peel flour in the bread (lines 263 to 320). One of the limitation factors about the percentage of peel added to the formulation is associated with the characteristic aroma and flavor of jabuticaba, which becomes very pronounced when the percentages increase. Unfortunately, we were unable to evaluate the aromas associated with jabuticaba peel flour in this paper.
Comments 4: Is the effect of Jabitica peel bloom during shelf life on the shelf life of packaging based on the influence of its own system pH on microorganisms or the generation of other functional components to exert antibacterial effects.
Response 4: We believe that the contribution of adding jabuticaba peel to the bread formulation may have increased the shelf life of the bread for several reasons, some of which are the modification of the pH, which may have positively influenced the fermentation process and the composition of the dough microbiota, as well as the metabolites formed during the fermentation process (lines 342 to 347). It would be interesting for future studies to focus on a more detailed analysis of the population of microorganisms in the dough and the metabolites formed.
Comments 5: The image quality or pixel count of the entire text is slightly low. It is recommended that the author provide high-quality images in the text. Suggest adding a picture of bread in the image to help reviewers or authors identify differences.
Response 5: The bread image was added to the manuscript as Figure 2.
Comments 6: In the conclusion, the author needs to elaborate on which beneficial qualities of bread can be improved by adding jabuticaba peel flower to bread.
Response 6: In the second paragraph of the conclusions, we point out to the reader that adding jabuticaba peel flour to the bread improved the glycemic response, which is important given that white bread has a high glycemic index.
Reviewer 2 Report
Comments and Suggestions for Authors
Dear sirs/madams,
thank you for submitting your research. I found it really VERY interesting and well conducted.
I just have to give you some minor corrections and a couple of advices to give.
1 - I found six self-citations of corresponding author's previous works. Five of them are related to previous researches about the same topic so it's ok for me, but one is not (9) and should be removed.
2- Figure 1 - The last step must be changed in "cut and storage" instead of "cut and estocation".
3- english language used is fine, yet some minor typos in the text should be corrected.
4- the research is really interesting but in my opinion it lacks an important part: a sensory assessment about taste and acceptance of the modifed bread with Jacuticaba peel that can give a complete view of the topic and could dramatically improve the relevance of the work (do not need to modify the work, it's just a personal consideration)
On the other side, i really want to compliment the research' staff for the excellent explaination of statistical treatments' description, with appropriate choices for "normally" and "not normally" distributed data. It's not a common thing to read usually.
Comments on the Quality of English LanguageEnglish is fine. Just some typos here and there in the text.
Author Response
Comments 1: I found six self-citations of corresponding author's previous works. Five of them are related to previous researches about the same topic so it's ok for me, but one is not (9) and should be removed.
Response 1: We appreciate your feedback and have removed this reference.
Comments 2: Figure 1 - The last step must be changed in "cut and storage" instead of "cut and estocation".
Response 2: We appreciate your feedback and added a new figure.
Comments 3: english language used is fine, yet some minor typos in the text should be corrected.
Response 3: We have double-checked the English throughout the manuscript.
Comments 4: the research is really interesting but in my opinion it lacks an important part: a sensory assessment about taste and acceptance of the modifed bread with Jacuticaba peel that can give a complete view of the topic and could dramatically improve the relevance of the work (do not need to modify the work, it's just a personal consideration)
Response 4: We fully agree with the reviewer. Unfortunately, the study in question was greatly impacted by the Covid-19 pandemic. We had planned the sensory analysis, but it was impossible to carry it out. One of the positive points is that the participants in the clinical trial reported that they liked the bread with the addition of jabuticaba.
Comments 5: On the other side, i really want to compliment the research' staff for the excellent explaination of statistical treatments' description, with appropriate choices for "normally" and "not normally" distributed data. It's not a common thing to read usually.
Response 5: We appreciate your care and dedication in reviewing our work.
Reviewer 3 Report
Comments and Suggestions for Authors
The manuscript discusses the preparation of fermentation bread with jabuticaba peel flour (JPF). The authors studied different formulations of the prepared bread and the impact on the blood glucose and insulin level during a performed clinical trial. The study should be of interest to the readership of the journal. However, it is required that the authors improve the manuscript, mainly in the Methodology section. Additional inclusion of discussions is also required in the Results and Discussions section for enhanced understanding of the reader.
I have the following comments to the authors:
1. Section 2 should be presented in two subsections 2.1 Materials and 2.2 Methods. Currently all information relevant to the Materials and the followed methodology is presented in one section and cannot be easily followed by the reader.
2. Relevant to lines 129-138, the explanation provided here on the methodology is not sufficient for the reader. Th authors should present the methods e.g. Measurement of moisture content, Lipid content, etc. in subsections and add a brief explanation of each followed protocol. The followed procedure is currently unclear and the authors only provided references with insufficient explanation.
3. Referring to section starting at line 215, the clinical trial should be presented as a separate subsection. Again the presentation of the followed methodology together with the rest in section 2 makes it difficult for the reader to follow and distinguish the presented information.
4. In Figure 1, what do authors mean by “estocation”? please explain.
5. Please remove the "*" sign in equations 1, 2, and 3 also in the main text relevant to the explained parameters. Since delta L,
delta a, and delta b are defined as (L-L*), (a-a*), and (b-b*) and the "*" sign commonly shows the reference point. The usage of the "*" sign for the measured parameters is confusing for the reader. In equation 3, delta* should be corrected to deltaE, please also correct it in the main text.
6. In line 284, “pH>=4.7” should be corrected to “pH <=4.7”.
7. In line 305, please explain the abbreviation "FCJ" in the main text.
Also in the caption of Figure 3, and footnote of Table 5, provide the complete term for ORAC.
8. In line 311, please explain briefly here which properties of the polyphenols may contribute to the extended shelf-life of the bread.
9. Referring to lines 324-326, please explain also add it to the main text which factor in JPF contributed to an increase in bread volume compared with calcium propionate.

Author Response
Comments 1: Section 2 should be presented in two subsections 2.1 Materials and 2.2 Methods. Currently all information relevant to the Materials and the followed methodology is presented in one section and cannot be easily followed by the reader.
Response 1: We appreciate your comment and have made the subdivisions in the Materials and Methods section. Only the standardized and widely used methodologies were not described in detail.
Comments 2: Relevant to lines 129-138, the explanation provided here on the methodology is not sufficient for the reader. Th authors should present the methods e.g. Measurement of moisture content, Lipid content, etc. in subsections and add a brief explanation of each followed protocol. The followed procedure is currently unclear and the authors only provided references with insufficient explanation.
Response 2: We based our work on other articles published in Foods, in which the authors do not detail the proximate composition analyses; also, the scientific community widely knows the AOAC standardized methodologies.
Comments 3: Referring to section starting at line 215, the clinical trial should be presented as a separate subsection. Again the presentation of the followed methodology together with the rest in section 2 makes it difficult for the reader to follow and distinguish the presented information.
Response 3: We appreciate your comment and have made the subdivisions in the Clinical Trial section.
Comments 4: In Figure 1, what do authors mean by “estocation”? please explain.
Response 4: We corrected the figure – “Storage”.
Comments 5: Please remove the "*" sign in equations 1, 2, and 3 also in the main text relevant to the explained parameters. Since delta L, delta a, and delta b are defined as (L-L*), (a-a*), and (b-b*) and the "*" sign commonly shows the reference point. The usage of the "*" sign for the measured parameters is confusing for the reader. In equation 3, delta* should be corrected to deltaE, please also correct it in the main text.
Response 5: Thank you for your comment, we have changed the text.
Comments 6: In line 284, “pH>=4.7” should be corrected to “pH <=4.7”.
Response 6: Thank you for your comment, we have changed the text.
Comments 7: In line 305, please explain the abbreviation "FCJ" in the main text. Also in the caption of Figure 3, and footnote of Table 5, provide the complete term for ORAC.
Response 7: Thank you for your comment, we have changed the text.
Comments 8: In line 311, please explain briefly here which properties of the polyphenols may contribute to the extended shelf-life of the bread.
Response 8: This explanation appears in lines 352 to 354.
Comments 9: Referring to lines 324-326, please explain also add it to the main text which factor in JPF contributed to an increase in bread volume compared with calcium propionate.
Response 9: We cannot say for sure, but the favoring of fermentation may have been the reason for the bread's volume development (lines 335 to 337).
Reviewer 4 Report
Comments and Suggestions for Authors
The manuscript titled " Production of long fermentation bread with jabuticaba peel flour added: technological, functional aspects and impact on glycemic and insulinemic response" falls within the scope of the Foods Journal by addressing a research topic of considerable interest. However, upon reading this article, several questions and concerns arise.
Comments:
Line 49: I find it somewhat inappropriate to mention the Holy Bible in a scientific article. To clarify, I am also a Christian, but I believe that religious references should not be highlighted in research within the STEM field.
Line 79: I kindly request the authors to verify reference no. 5, as it appears that this article does not pertain to sourdough technology.
Line 99: The authors should provide a reference to support this statement.
Lines 115-118: This sentence should be removed from the Material & Methods section. It would be more appropriate to mention this later in the Discussion section.
Line 122: How were the bread samples stored for the staling analysis? Were they packed in foil or a similar material? How long were they cooled before packing?
Lines 147-148: “The pieces of bread were evaluated…”. Which pieces of bread are you referring to? I assume you mean the bread crumb? Method 44-15.02 refers to method 62-05.01, which prescribes slicing the entire loaf of bread, not just the crumb. Why did you choose to evaluate only the crumb?
Line 150: The AACC method 72-10 (Specific Volume) was deleted in 2000. Please refer to the latest method (10-05) if the rapeseed displacement was used.
Line 152: The AACC method 72-10.02 is not relevant for bread firmness analysis. Which equipment was used for the texture analysis?
Line 156: Results for the hue angle parameter are not presented in the Results and Discussion section. Is there a specific reason for this? Additionally, I believe the hue angle is not indicated with an asterisk (h*), but possibly with a degree (h°) or simply with the letter (h). Please verify this.
Line 165: The letter “E” is missing.
Line 206: Was only the bread crumb used for the microbiological analyses, or was the crust also included?
Lines 229-230: “based on providing 50 g of available carbohydrates (112 g of JPF2 bread and 128 g of control bread).” I find the indicated amounts of bread confusing. As shown in Table 2, sample JPF2 has fewer carbohydrates (40.83%) than the control bread (42.47%). Therefore, it would be logical to conclude that a portion containing 50 g of carbohydrates would require more JPF2 sample than the control bread. According to my calculations, it would be necessary to prepare 122 g (50/40.83x100) of the JPF2 sample and 118 g (50/42.47x100) of the control sample to provide an equal amount of 50 g of carbohydrates. If my suspicions are correct, this may call into question the overall results of the clinical trials.
Lines 306-308: “Firmness … was more significant in bread with JPF since mold development was observed in control bread from the fifth day of storage.” The relationship between firmness and mould is unclear to me. Please rephrase this sentence to make it more comprehensible.
Line 310: Similar to the previous point. “JPF in the formulation also acted as a natural preservative in the bread due to the presence of polyphenols”. Authors, as well as readers, should understand that bread staling and bread spoilage are two distinct and entirely different processes.
Lines 319-321: According to Table 3, the specific volume of the control bread was significantly higher than that of the JPF samples. Please emphasise this in the text.
Lines 343-347: In a study conducted by Rinaldi et al. (2017), I could not find information regarding the influence of pH on bread preservation. Please verify this and confirm the statement that pH 4.5 of bread is the threshold above which bread can be preserved, and that a pH lower than 4.7 is the value below which bread can be preserved for an extended period. Based on the results of your research, it cannot be concluded that pH 4.5-4.7 is the definitive threshold for bread preservation, as other factors (e.g. polyphenols) may also influence preservation. Please revise this paragraph.
Line 348 (Table 3): If available, present the water activity with at least two decimal places.
Line 351: “Different capital letters in the same row or different lowercase letters in the columns represent statistical differences between samples.” For instance, it is evident that lowercase letters refer to rows (not columns) in the case of pH, texture, colour, and volume. I am unsure if this is correct (except perhaps for moisture). Please verify the entire table.
Line 360 (Table 4): Table 4 requires a more extensive discussion. Please provide a more thorough commentary on the results from Table 4.
Line 402 (Table 5): What does the last row (JPF) in Table 5 refer to? Why was the statistical analysis of significant differences not performed for the parameters IAA and TRC?
Line 406: “Different means with lowercase letters in the lines differ significantly from each other according to Tukey's test (p<0.05).” I believe this is incorrect.
Line 412: Please explain in greater detail why the JPF2 sample was considered the most suitable for the clinical trial.
Lines 429-442: Please support your claims with references.
Lines 475-479: Please support your claims with references.
Lines 418-530 (3.4. Clinical trial): The results presented in this section may be called into question due to the experimental design flaws (incorrectly calculated sample doses). The authors should address these concerns in detail.
Author Response
Comments 1: Line 49: I find it somewhat inappropriate to mention the Holy Bible in a scientific article. To clarify, I am also a Christian, but I believe that religious references should not be highlighted in research within the STEM field.
Response 1: We understand the reviewer's position and have removed the mention of the Holy Bible from the text.
Comments 2: Line 79: I kindly request the authors to verify reference no. 5, as it appears that this article does not pertain to sourdough technology.
Response 2: Thanks for your comment. We have replaced this reference.
Comments 3: Line 99: The authors should provide a reference to support this statement.
Response 3: Thanks for your comment. We have added reference.
Comments 4: Lines 115-118: This sentence should be removed from the Material & Methods section. It would be more appropriate to mention this later in the Discussion section.
Response 4: Thank you for your comment; we have inserted this phrase into the results and discussion.
Comments 5: Line 122: How were the bread samples stored for the staling analysis? Were they packed in foil or a similar material? How long were they cooled before packing?
Response 5: This information was included in Figure 1 and in the text.
Comments 6: Lines 147-148: “The pieces of bread were evaluated…”. Which pieces of bread are you referring to? I assume you mean the bread crumb? Method 44-15.02 refers to method 62-05.01, which prescribes slicing the entire loaf of bread, not just the crumb. Why did you choose to evaluate only the crumb?
Response 6: We have corrected the information in the manuscript.
Comments 7: Line 150: The AACC method 72-10 (Specific Volume) was deleted in 2000. Please refer to the latest method (10-05) if the rapeseed displacement was used.
Response 7: Thanks for your comment. We have updated the reference.
Comments 8: Line 152: The AACC method 72-10.02 is not relevant for bread firmness analysis. Which equipment was used for the texture analysis?
Response 8: Thanks for your comment; we have added this information.
Comments 9: Line 156: Results for the hue angle parameter are not presented in the Results and Discussion section. Is there a specific reason for this? Additionally, I believe the hue angle is not indicated with an asterisk (h*), but possibly with a degree (h°) or simply with the letter (h). Please verify this.
Response 9: Thanks for your comment; we have corrected this information.
Comments 10: Line 165: The letter “E” is missing.
Response 10: Thanks for your comment; we have corrected this information.
Comments 11: Line 206: Was only the bread crumb used for the microbiological analyses, or was the crust also included?
Response 11: Thanks for your comment; we have corrected this information.
Comments 12: Lines 229-230: “based on providing 50 g of available carbohydrates (112 g of JPF2 bread and 128 g of control bread).” I find the indicated amounts of bread confusing. As shown in Table 2, sample JPF2 has fewer carbohydrates (40.83%) than the control bread (42.47%). Therefore, it would be logical to conclude that a portion containing 50 g of carbohydrates would require more JPF2 sample than the control bread. According to my calculations, it would be necessary to prepare 122 g (50/40.83x100) of the JPF2 sample and 118 g (50/42.47x100) of the control sample to provide an equal amount of 50 g of carbohydrates. If my suspicions are correct, this may call into question the overall results of the clinical trials.
Response 12: To define the clinical trial portion size, we only consider available carbohydrates. The calculations are not based on total carbohydrates. In addition, we must consider that jabuticaba peel flour also contributes free sugars to the formulation. We have added information to the text regarding the methodology used to define available carbohydrates.
Comments 13: Lines 306-308: “Firmness … was more significant in bread with JPF since mold development was observed in control bread from the fifth day of storage.” The relationship between firmness and mould is unclear to me. Please rephrase this sentence to make it more comprehensible.
Response 13: We were able to evaluate the texture of the bread with JPF up to the 7th day, with the values ​​increasing. In the control bread, due to mold formation from the 5th day onwards, it was impossible to continue the evaluation.
Comments 14: Line 310: Similar to the previous point. “JPF in the formulation also acted as a natural preservative in the bread due to the presence of polyphenols”. Authors, as well as readers, should understand that bread staling and bread spoilage are two distinct and entirely different processes.
Response 14: We agree with you, and we have included additional information in this phrase.
Comments 15: Lines 319-321: According to Table 3, the specific volume of the control bread was significantly higher than that of the JPF samples. Please emphasise this in the text.
Response 15: This information is in line 335.
Comments 16: Lines 343-347: In a study conducted by Rinaldi et al. (2017), I could not find information regarding the influence of pH on bread preservation. Please verify this and confirm the statement that pH 4.5 of bread is the threshold above which bread can be preserved, and that a pH lower than 4.7 is the value below which bread can be preserved for an extended period. Based on the results of your research, it cannot be concluded that pH 4.5-4.7 is the definitive threshold for bread preservation, as other factors (e.g. polyphenols) may also influence preservation. Please revise this paragraph.
Response 16: In fact, a combination of factors increased bread's shelf life (water activity, pH, phenolics). We added one more sentence to the discussion to clarify this relationship.
Comments 17: Line 348 (Table 3): If available, present the water activity with at least two decimal places.
Response 17: We have corrected this information.
Comments 18: Line 351: “Different capital letters in the same row or different lowercase letters in the columns represent statistical differences between samples.” For instance, it is evident that lowercase letters refer to rows (not columns) in the case of pH, texture, colour, and volume. I am unsure if this is correct (except perhaps for moisture). Please verify the entire table.
Response 18: We have corrected this information.
Comments 19: Line 360 (Table 4): Table 4 requires a more extensive discussion. Please provide a more thorough commentary on the results from Table 4.
Response 19: Microbiological analyses were performed mainly to comply with legislation and ensure that food was safe for use in clinical trials. This information has been added to the text.
Comments 20: Line 402 (Table 5): What does the last row (JPF) in Table 5 refer to? Why was the statistical analysis of significant differences not performed for the parameters IAA and TRC?
Response 20: It was impossible to perform statistical analysis on these indexes, as they were calculated using average ORAC and TRC data.
Comments 21: Line 406: “Different means with lowercase letters in the lines differ significantly from each other according to Tukey's test (p<0.05).” I believe this is incorrect.
Response 21: We have corrected this information.
Comments 22: Line 412: Please explain in greater detail why the JPF2 sample was considered the most suitable for the clinical trial.
Response 22: Thanks for your comment; we have added this information.
Comments 23: Lines 429-442: Please support your claims with references.
Response 23: We have added reference.
Comments 24: Lines 475-479: Please support your claims with references.
Response 24: We have added reference.
Comments 25: Lines 418-530 (3.4. Clinical trial): The results presented in this section may be called into question due to the experimental design flaws (incorrectly calculated sample doses). The authors should address these concerns in detail.
Response 25: To define the clinical trial portion size, we only consider available carbohydrates. The calculations are not based on total carbohydrates. In addition, we must consider that jabuticaba peel flour also contributes free sugars to the formulation. We have added information to the text regarding the methodology used to define available carbohydrates.
Round 2
Reviewer 1 Report
Comments and Suggestions for Authors
The article can be considered accepted
Reviewer 4 Report
Comments and Suggestions for Authors
The authors accepted all my suggestions for corrections.